# Binary Reward Labeling: Bridging Offline Preference and Reward-Based Reinforcement Learning

## Abstract

Offline reinforcement learning has become one of the most practical RL settings. However, most existing works on offline RL focus on the standard setting with scalar reward feedback. It remains unknown how to universally transfer the existing rich understanding of offline RL from the reward-based to the preference-based setting. In this work, we propose a general framework to bridge this gap. Our key insight is transforming preference feedback to scalar rewards via binary reward labeling (BRL), and then any reward-based offline RL algorithms can be applied to the dataset with the reward labels. The information loss during the feedback signal transition is minimized with binary reward labeling in the practical learning scenarios. We theoretically show the connection between several recent PBRL techniques and our framework combined with specific offline RL algorithms. By combining reward labeling with different algorithms, our framework can lead to new and potentially more efficient offline PBRL algorithms. We empirically test our framework on preference datasets based on the standard D4RL benchmark. When combined with a variety of efficient reward-based offline RL algorithms, the learning result achieved under our framework is comparable to training the same algorithm on the dataset with actual rewards in many cases and better than the recent PBRL baselines in most cases.

## 1 Introduction

Reinforcement learning (RL) is an important learning paradigm for solving sequential decision-making problems (Sutton & Barto, 2018). Compared to the standard (reward-based) RL that requires access to reward feedback (Schulman et al., 2017), preference-based RL (PBRL) (Wirth et al., 2017) only requires preference feedback over a pair of trajectories, making it more accessible in practice. In the offline learning setting, the agent only needs a pre-collected dataset of preference labels before training, making it even more convenient (Zhu et al., 2023). When humans provide preference labels, PBRL is known as reinforcement learning from human feedback (RLHF) (Stiennon et al., 2020), a popular framework for aligning large language models.

Despite the success in offline PBRL (Kim et al., 2023; Dai et al., 2023; Hejna & Sadigh, 2024), the topic is less studied compared to the offline RL in the standard reward feedback setting (Li et al., 2024; Levine et al., 2020; Fujimoto & Gu, 2021; Kidambi et al., 2020; Wu et al., 2019; Cheng et al., 2022; Tarasov et al., 2024). The critical feature in offline learning is 'pessimism' (Li et al., 2024). The learning agent should be pessimistic about the policies whose behaviors are not included in the dataset. Many pessimistic learning algorithms with theoretical insights have been developed for the standard offline RL setting. However, most empirical works for offline PBRL only adopt limited specific pessimistic learning approaches, such as applying certain regularization to the learning policy (Rafailov et al., 2024; Stiennon et al., 2020), which are shown to be less efficient compared to recent SOTA methods in the standard RL setting (Cheng et al., 2022; Tarasov et al., 2024).

The difference between the current PBRL and standard RL only comes from the feedback signal's format. Both settings assume the environment as a Markov decision process (MDP) and aim to find the policy that achieves a high cumulative reward (i.e., is aligned with the reward function). The

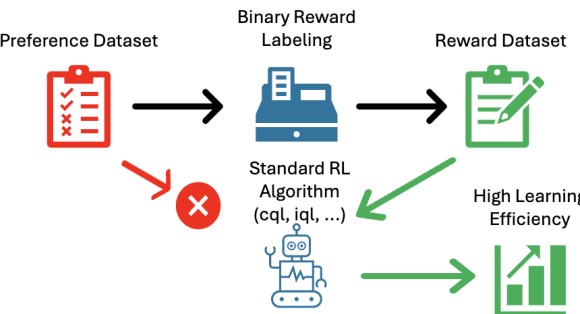

Figure 1: Illustration of the binary reward labeling (BRL) framework. Given a dataset consisting of preference feedback as input, the BRL framework labels the dataset with the rewards that best explain the preference signals from the dataset. With a dataset of reward labels, the learning agent can train on the dataset with any efficient offline RL algorithms such as IQL (Kostrikov et al., 2021) or CQL (Kumar et al., 2020) and enjoy high learning efficiency.

question arises whether one can utilize the existing well-developed standard offline RL algorithms to solve the PBRL problem. Wang et al. (2024) has theoretically demonstrated this feasibility in online learning under certain conditions. This work aims to develop a framework that bridges the gap between PBRL and standard RL so that one can solve a PBRL problem with a standard offline RL algorithm. To construct such a framework, one faces the following difficulties:

- Standard offline RL algorithms are very different from each other. Model-based algorithms (Kidambi et al., 2020; Yu et al., 2020) learn an environment model and then run online learning algorithms on the model. Model-free algorithms (Cheng et al., 2022; Fujimoto & Gu, 2021) learn a pair of actor-critic models directly to infer the optimal policy without modeling the environment. The framework needs to apply to all kinds of learning algorithms.

- Preference feedback contains arguably less information than reward feedback. It is impractical to recover the actual rewards from limited preference feedback. The framework needs to make the standard offline RL algorithms work with incomplete information from preference feedback.

In this work, we build a novel framework called 'BRL' based on a reward labeling technique that labels the preference dataset with scalar rewards. The critical insight is that we only need to consider transforming feedback signals from preference to reward format while minimizing information loss. The standard offline RL algorithm can handle the pessimistic learning afterward. Note that reward labeling fundamentally differs from reward modeling, which is widely considered in recent PBRL studies (Kim et al., 2023; Stiennon et al., 2020). Reward modeling aims to infer rewards at state actions that are not in the dataset. In contrast, reward labeling focuses on interpreting the rewards for the state actions inside the preference dataset. To minimize the information loss in the transformation, we want to find the optimal reward labels that best explain the state actions in the dataset. We show that a simple binary labeling technique already gives the optimal reward labels in a practical case where the trajectories in the dataset have no overlap. We further empirically show that in the general case where no matter whether there is an overlap between trajectories or not, the learning algorithms achieve better performance on the reward labels given by the binary labeling technique than on those given by the reward modeling technique. In Fig 1, we illustrate how our framework is used. In summary, our contributions are as follows.

- We propose a general framework to bridge the gap between offline PBRL and standard offline RL. Our framework involves labeling the dataset with the reward that maintains most information in the preference signals. Afterward, one can train on the reward-labeled dataset with any standard offline RL algorithms. Our framework can be easily implemented without modifying the original RL algorithms. One can possibly construct more efficient PBRL algorithms by applying SOTA standard RL algorithms to our framework.

- We mathematically analyze the combination of our framework with standard offline RL algorithms. We show that current state-of-the-art PBRL techniques are closely related to the combi-

nation of our framework and some specific standard offline RL algorithms regarding utilizing the preference labels.

- We empirically show that our method performs significantly better on standard evaluation benchmarks than existing SOTA methods. In many cases, our method can even compete with training the same RL algorithm on the corresponding dataset of true reward labels.

## 2 RELATED WORKS

### 2.1 OFFLINE REWARD-BASED REINFORCEMENT LEARNING

Offline reward-based reinforcement learning, or standard offline RL, is the most popular offline RL setting. The fundamental challenge for offline RL is known as 'distribution mismatch' (Levine et al., 2020). It refers to the phenomenon that the data distribution in the dataset may not match the distribution induced by the optimal policy. Due to distribution mismatch, an efficient learner must be 'pessimistic' (Li et al., 2024). In general, it says that an agent should rely more on the policies whose distributions match the dataset distribution better, as the agent cannot correctly evaluate other policies not covered by the dataset. Many algorithms have been proposed to solve the problem following very different pessimistic learning techniques. Kidambi et al. (2020); Yu et al. (2020; 2021) proposed model-based learning algorithms that learn a world model first and then apply online RL to learn from the world model. Wu et al. (2019) proposes a behavior regularization approach closely related to the regularization considered in the RLHF studies. Fujimoto & Gu (2021); Kostrikov et al. (2021); Kumar et al. (2020); Levine et al. (2020) proposed model-free learning algorithms that don't need to learn the world model. Cheng et al. (2022) provide theoretical guarantees on the efficiency of their model-free learning methods that the learned policy is always better than the behavior policy used to collect the dataset and can compete with the best policy covered by the dataset. Li et al. (2024) mathematically interprets the essence of pessimistic learning algorithms.

### 2.2 OFFLINE PREFERENCE-BASED REINFORCEMENT LEARNING

Zhan et al. (2023); Zhu et al. (2023) theoretically investigate the problem of pessimistic learning in the standard offline PBRL setting, the same as the one considered in this work. They propose a method that is guaranteed to learn near-optimal policy depending on the dataset, but it remains unknown how to implement these algorithms in practice. Stiennon et al. (2020); Ouyang et al. (2022) studies the problem of language model fine-tuning, where humans provide the preference labels. These works, also known as RLHF, do not apply to the general RL setting. These works also only consider behavior regularization for pessimistic learning, which is shown to be less efficient than other advanced pessimistic learning techniques.

A line of research studies a variant of the offline learning setting. Here, the learning agent can access a preference dataset consisting of preference labels over trajectory pairs and a demonstration dataset consisting of only trajectories. Kim et al. (2023) proposes to use a transformer architecture to approximate the reward model. Hejna & Sadigh (2024) proposes to adapt the IQL learning framework to the preference-based learning setting. Zhang et al. (2023) proposes to learn a preference model instead of a reward model and then learn a policy aligned with the preference model.

Another line of research Sadigh et al. (2017); Shin et al. (2021; 2023) studies the PBRL problem in the active learning setting. In this case, the learning starts from a pre-collected behavior dataset with no preference label. During training, the agent can query an expert to provide preference feedback on a pair of trajectories sampled from the behavior dataset by the agent in an online manner. Then, the agent learns a reward model from the preference feedback and applies RL algorithms to learn from the reward model.

## 3 PRELIMINARIES

### 3.1 REINFORCEMENT LEARNING

First, we introduce the general RL framework, where an agent interacts with an environment at discrete time steps. The environment is characterized by a Markov Decision Process (MDP) $\mathcal{M} =$

$\{\mathcal{S}, \mathcal{A}, \mathcal{R}, \mathcal{D}\}$ where $\mathcal{S}$ is the state space, $\mathcal{A}$ is the action space, $\mathcal{R}$ is the reward function, and $\mathcal{D}$ is the state transition dynamics. At each time-step, the environment is at a state $s \in \mathcal{S}$, and the agent takes an action $a \in \mathcal{A}$, and then the environment transits to the next state $s' \sim \mathcal{D}(\cdot|s, a)$ with an instant reward $r = R(s, a)$. Without loss of generality, we assume the rewards are bounded in $[-1, 1]$. A policy $\pi : \mathcal{S} \rightarrow \Delta(\mathcal{A})$ represents a way to interact with the environment by sampling an action from the distribution given by $\pi(s)$ at a state $s$. Given an initial state $s_0$, the performance of a policy is evaluated through its discounted cumulative reward $J(\pi) = \sum_{t=0}^{\infty} \mathbb{E}[\gamma^t R(s^t, a^t)|a^t \sim \pi(s^t)]$, where $\gamma$ is a discount factor. In the standard RL framework, the learner can observe scalar reward feedback for a state action, which is not the case for the preference-based setting.

## 3.2 Preference Based Reinforcement Learning (PBRL)

PBRL is a variant of RL where the feedback signals consist of preferences rather than scalar rewards. Given a pair of sequences of states and actions, also known as trajectories, a preference model $P$ gives a preference for the trajectories. Formally, let $\mathcal{T} = (\mathcal{S}, \mathcal{A})^T$ be the space of trajectories with length $T$, the preference model is a mapping $P : \mathcal{T} \times \mathcal{T} \rightarrow [0, 1]$. $P(\tau_1, \tau_2)$ represents the probability of the preference model preferring the first trajectory $\tau_1$ over the second one $\tau_2$. Following recent PBRL studies, we assume the preference model is related to the reward function. More specifically, there exists a monotonically increasing link function $f : \mathbb{R} \rightarrow [0, 1]$ bounded in $[0, 1]$ such that $P(\tau_1, \tau_2) = f(\sum_{(s,a)\in\tau_1} R(s, a) - \sum_{(s,a)\in\tau_2} R(s, a))$. The popular Bradley-Terry model, considered in many recent works, uses the sigmoid function as the link function. Our method requires no specific knowledge of the link function in this work.

## 3.3 Offline PBRL Setting

This work studies PBRL in the standard offline setting Zhan et al. (2023). Here, the environment is still characterized by an MDP $\mathcal{M} = \{\mathcal{S}, \mathcal{A}, \mathcal{R}, \mathcal{D}\}$, but the learner cannot directly interact with the environment or observe any reward signals from $\mathcal{R}$. The learner knows the state and action spaces $\mathcal{S}, \mathcal{A}$ and can access a pre-collected dataset $D$ of preference feedback. The dataset $D$ contains $N$ tuples of data, and each tuple consists of a pair of trajectories and a preference label. The preference labels in the dataset are generated by a preference model $P$ unknown to the learning agent. For each pair of trajectory $(\tau_1, \tau_2)$, the preference model randomly generates a preference signals $\sigma \in \{1, 2\}$ through a Bernoulli trial with a probability $p = P(\tau_1, \tau_2)$. The preference signal $\sigma$ represents the index of the preferred trajectory. For convenience, we call the preferred trajectory 'the chosen trajectory' and the other trajectory 'the rejected trajectory.' Wang et al. (2023) considers a different setting where multiple preference labels are generated for each pair of trajectories, and we will extend our main method to this setting in the experiments. Note that the preference model $P$ is determined by a link function $f$ and the reward function $\mathcal{R}$ of the environment, and both should be unknown to the learner in principle. It is common among recent works that assume $f$ to be a sigmoid function, but in this work, our method does not require the knowledge of $f$. At a high level, the agent's goal is to reliably learn a policy from the preference dataset $D$ with a high cumulative reward based on the underlying reward function $\mathcal{R}$.

# 4 Bridging Preference and Reward-Based Offline Reinforcement Learning

To utilize existing efficient reward-based RL algorithms for offline PBRL, we consider an information-translation approach that is applicable to universal reward-based RL algorithms. By assigning a reward label for each state-action in the dataset, any reward-based RL algorithm can be trained on the new dataset with the reward labels. Therefore, our task is to find the reward labels that contain the closest information to the preference labels. Note that it is a common approach to train a reward model (also known as reward modeling) and then use the reward model to generate the reward labels Christiano et al. (2017), but this is not necessary in our case as we only need to assign reward labels to the state-actions in the dataset.

## 4.1 Optimal Reward Labeling

First, we introduce the basic metric to evaluate how well the rewards labels interpret the preference labels. Given an offline preference dataset $\mathcal{D} = \{(\tau_1^i, \tau_2^i, \sigma^i)\}, i \in [N]$. Without loss of generality, we assume $\sigma^i \equiv 1$. That is, the first trajectory in each pair is always the chosen trajectory, and the second ones are always the rejected trajectories. In this case, we simplify the notation of the dataset as $\mathcal{D} = \{\tau_1^i \succ \tau_2^i\}, i \in [N]$. Let $(s_{j,t}^i, a_{j,t}^i), j \in [2], t \in [T]$ be the $t^{th}$ state-action pairs of trajectory $j$ from the $i^{th}$ data tuple, and let $r_{j,t}^i$ be the reward label for this state action pair. In addition, the reward labels should be generated using the same reward model. In other words, there exists a reward function $\hat{\mathcal{R}}$, such that $r_{j,t}^i = \hat{\mathcal{R}}(s_{j,t}^i, a_{j,t}^i)$. Let $f : \mathbb{R} \to [0, 1]$ be the monotonically increasing link function between the rewards and the preference, the probability of preferring the chosen trajectory $\tau_1^i$ over $\tau_2^i$ predicted by the rewards label is $p = f(\sum_t r_{1,t}^i - \sum_t r_{2,t}^i)$. Then, a monotonically decreasing loss function $L : [0, 1] \to \mathbb{R}$ can be defined based on the probability that the rewards predict the chosen trajectory. One can use the total prediction loss on preference to evaluate the quality of the reward labels. We denote $F(\cdot) = L(f(\cdot))$ as the link-loss function to represent the combination. Note that the loss and link functions are monotonically decreasing and increasing, respectively. So, the link-loss function is monotonically decreasing. This link-loss function has been widely considered in recent PBRL studies that consider reward modeling. For reward modeling, the reward labels in the loss function are replaced by the reward predictions by the reward model at the corresponding state actions. In these works, the common choice for $f$ is the sigmoid function, and $L$ is the negative log of the KL divergence Christiano et al. (2017) .

The reward labels with minimal prediction loss have information closest to the preference dataset among all possible reward labels. Formally, the optimal reward labels are defined as follows.

**Definition 4.1** (Optimal reward label). Given a preference dataset $D = \{(\tau_1^i \succ \tau_2^i)\}, i \in [N]$. Let $\mathbf{r}$ be the reward labels for the dataset, and $r_{j,t}^i$ be the reward label for the state-action pair of trajectory $j$ at step $t$ of the $i^{\text{th}}$ data tuple. Let $F : \mathbb{R} \to \mathbb{R}$ be a link-loss function to represent the prediction loss of a reward label on a preference label. The optimal reward labels for this dataset are the solution to the optimization problem as below:

$$\arg\min_r \sum_{i \in [N]} F(\sum_{t \in [T]} r_{1,t}^i - \sum_{t \in [T]} r_{2,t}^i))$$

$$s.t. \exists \hat{\mathcal{R}} : \mathcal{S} \times \mathcal{A} \to [0, 1], r_{j,t}^i = \hat{\mathcal{R}}(s_{j,t}^i, a_{j,t}^i). \tag{1}$$

Here $(s_{j,t}^i, a_{j,t}^i), j \in [2], t \in [T]$ are the $t^{th}$ state-action pairs of trajectory $j$ from the $i^{th}$ data tuple

To minimize the information loss during the feedback signal transition, one should use the optimal reward to re-label the dataset. In general cases, finding the exact solution can be complicated, and it is common to use a deep neural network to approximate the reward function Christiano et al. (2017) that minimizes the prediction loss. However, in practice, the same state-action pairs will usually not appear multiple times in the dataset. For example, in RLHF, each prompt usually samples two different answers to label (Touvron et al., 2023); in continuous control, the state and action spaces are continuous, making it unlikely to visit the same state action twice. Note that the reward model constraint requires the reward labels for the same state-action to be the same, and this makes no difference if each state-action is unique in the dataset. In this case, we can directly derive the exact solution to the optimal reward labels as shown in Lemma 4.2 below.

**Lemma 4.2.** *Consider a preference dataset $D = \{(\tau_1^i \succ \tau_2^i)\}, i \in [N]$. If each state-action pair is unique in the dataset, the optimal reward labels are:*

$$r_{j,t}^i = \begin{cases} +1, & \forall t \in [T], \forall i \in [N], if\, j = 1 \\ -1, & \forall t \in [T], \forall i \in [N], if\, j = 2 \end{cases} \tag{2}$$

The proof for Lemma 4.2 is in the Appendix. Lemma 4.2 says that in the typical cases where there is no overlap between the trajectories, the optimal reward labels are binary signals: for all state actions from the chosen trajectories, the optimal reward labels are the maximal rewards; for all state actions from the rejected trajectories, the optimal reward labels are the minimal rewards. Formally, Alg 1 shows how to apply an arbitrary offline reward-based RL algorithm to our framework.

---

**Algorithm 1:** Binary Reward Labeling for Offline PBRL

---

**Input:** preference dataset, offline reward-based RL algorithm Alg
1. Label the state-actions in the dataset with the optimal reward labels according to Eq 2. The resulting dataset with reward labels is $\mathcal{D}_R$.
2. Run algorithm Alg on the reward dataset $\mathcal{D}_R$ to learn a policy $\pi$.
**Output:** $\pi$.

---

Next, we discuss the general case where the same state action appears in different chosen and rejected trajectories. As mentioned earlier, a common approach is to approximate the solution through reward modeling with a deep neural network. Alternatively, one can still apply the binary labeling technique from Lemma 4.2, and the labels for the repeated state actions in effect are the mean of the binary rewards. This is only a sub-optimal solution. However, in Section 5, our empirical results show that such a simple binary labeling method is efficient in both overlapped and no overlap cases and is generally more efficient than the reward modeling method. Note that another advantage of BRL compared to reward modeling is that it does not require learning a reward model.

### 4.2 THEORETICAL ANALYSIS

In this section, we show that existing PBRL techniques are closely related to some special cases of combining specific offline RL algorithms with our framework. Under certain conditions, they can even be equivalent.

**Offline standard RL algorithms are model-based.** First, we consider the case of model-based algorithms. Usually, a model-based offline RL algorithm utilizes the reward signals to learn the reward model of the environment (Yu et al., 2020; 2021). Formally, we characterize the reward modeling process in a general model-based offline RL algorithm in Definition 4.3.

**Definition 4.3.** (reward modeling in model-based approaches) Given a reward-based dataset $\mathcal{D}_{\mathcal{R}} = \{(s_i, a_i, r_i), i \in [N]\}$, the reward modeling process is solving the optimization problem

$$\min_{\widehat{\mathcal{R}}} \sum_{(r,s,a) \in D} |\widehat{\mathcal{R}}(s,a) - r|$$

, where $\widehat{\mathcal{R}} : \mathcal{S} \times \mathcal{A} \to [-1, 1]$ is a reward model.

Here, we extend Definition 4.3 to the case of the preference dataset with binary reward labeling. Given a preference dataset $\mathcal{D} = \{(\tau_1^i \succ \tau_2^i)\}, i \in [N]$, after applying the binary reward labeling through Alg 1, the reward-based dataset $\mathcal{D}_{\mathcal{R}}$ consists of all state-actions from $\mathcal{D}$. For each state-action $(s, a) \in \mathcal{D}_{\mathcal{R}}$, the reward label is $+1$ if the state-action comes from a chosen trajectory and $-1$ otherwise. In this case, the optimization problem becomes

$$\min_{\widehat{\mathcal{R}}} \sum_{i \in [N]} \sum_{(s,a) \in \tau_1^i} |1 - \widehat{\mathcal{R}}(s,a)| + \sum_{(s,a) \in \tau_2^i} |\widehat{\mathcal{R}}(s,a) + 1| := \sum_{i \in [N]} \min_{\widehat{\mathcal{R}}} \mathcal{L}_1(\tau_1^i, \tau_2^i, \mathcal{R}).$$

Next, we formally introduce the reward modeling process based on the preference dataset directly in Definition 4.4. As we explained earlier, this process is similar to solving the optimal reward label problem in Definition 4.1 and standard in current PBRL studies.

**Definition 4.4.** (reward modeling on preference signals) Given a preference-based dataset $\mathcal{D} = \{(\tau_1^i \succ \tau_2^i)\}, i \in [N]$, the reward modeling process is solving the optimization problem below:

$$\min_{\widehat{\mathcal{R}}} \sum_{i \in [N]} F\Big( \sum_{(s,a) \in \tau_1^i} \hat{R}(s,a) - \sum_{(s,a) \in \tau_2^i} \hat{R}(s,a) \Big) := \sum_{i \in [N]} \min_{\widehat{\mathcal{R}}} \mathcal{L}_2(\tau_1^i, \tau_2^i, \mathcal{R}).$$

Finally, in Theorem 4.5, we formally show the connection between the two methods.

**Theorem 4.5.** *Given a preference dataset $\mathcal{D} = \{(\tau_1^i \succ \tau_2^i)\}, i \in [N]$, reward modeling can be performed either on the preference dataset directly as Definition 4.4 or on the reward dataset as Definition 4.3 with the dataset generated binary reward labeling through Alg 1. The two methods are connected in the following three cases:*

1. *When there is no overlap between the trajectories in the dataset, the optimal solutions in both methods are the same:* $\arg\min_{\widehat{\mathcal{R}}} \sum_{i \in [N]} \mathcal{L}_1(\tau_1^i, \tau_2^i, \widehat{\mathcal{R}}) = \arg\min_{\widehat{\mathcal{R}}} \sum_{i \in [N]} \mathcal{L}_2(\tau_1^i, \tau_2^i, \widehat{\mathcal{R}})$

2. *If the link-loss function $\mathcal{F}$ is linear, then the optimization problems in both methods are equivalent:* $\sum_{i \in [N]} \mathcal{L}_1(\tau_1^i, \tau_2^i, \widehat{\mathcal{R}}) = C_1 \cdot \sum_{i \in [N]} \mathcal{L}_2(\tau_1^i, \tau_2^i, \widehat{\mathcal{R}}) + C_2$, *where $C_1, C_2$ are constant scalars.*

3. *Let $w$ be the parameter of the reward function $\mathcal{R}$. For each trajectory pair, the gradients of its contribution to the optimization goal on the reward function parameter have the same direction in the two methods:* $\frac{\partial \mathcal{L}_1(\tau_1^i, \tau_2^i, \widehat{\mathcal{R}})}{\partial w} / \| \frac{\partial \mathcal{L}_1(\tau_1^i, \tau_2^i, \widehat{\mathcal{R}})}{\partial w} \| = \frac{\partial \mathcal{L}_2(\tau_1^i, \tau_2^i, \widehat{\mathcal{R}})}{\partial w} / \| \frac{\partial \mathcal{L}_2(\tau_1^i, \tau_2^i, \widehat{\mathcal{R}})}{\partial w} \|.$

The proof for Theorem 4.5 can be found in the appendix. The first case shows that in the most practical scenario where there is no overlap between trajectories, the reward modeling in both methods approximates the same optimal reward function. The second case shows that the reward modeling in both methods is equivalent if the link-loss function is linear. The third case shows that the reward model update based on each trajectory pair in both methods is the same. In conclusion, the way how a model-based algorithm utilizes the preference signals for reward modeling under our framework is closely related to that of the current reward-modeling PBRL approaches.

**Offline standard RL algorithms are model-free.**

In the case where our framework is combined with a reward-based algorithm, we find a similar result. One can adapt a standard model-free offline RL algorithm to the PBRL setting with some intuitive techniques as in Hejna & Sadigh (2024). We show that such methods utilize the preference labels in the same way as our framework when combined with the same model-free algorithm under the condition that the link-loss function $F$ is linear. The detailed analysis is in the Appendix.

## 5 EXPERIMENTS

### 5.1 EXPERIMENTS SETUP

For the construction of the preference dataset, we sample pairs of trajectories from the offline RL benchmark D4RL (Fu et al., 2020) and generate synthetic preference following the standard techniques in previous PBRL studies (Kim et al., 2023; Christiano et al., 2017). Formally, we introduce the process as follows.

1. Randomly sample pairs of trajectory clips from the original D4RL dataset. The length of the clip is set to be 20 steps, which is similar to previous studies (Christiano et al., 2017).

2. For each pair of trajectory clips, find the probability of a trajectory to be preferred by applying their regularized rewards in the D4RL datasets to the Bradley-Terry model. The regularized rewards are bound in $[-1, 1]$ to ensure consistency between different datasets.

3. For each pair of trajectory clips, generate a preference label through a Bernoulli trial with the probability from the second step.

4. Return the preference dataset consisting of the trajectory clip pairs and the corresponding preference labels.

To ensure that different types of datasets are covered, we chose D4RL datasets from different environments, including HalfCheetah, Walker2d, and Hopper, with different types of trajectories, including medium, medium-expert, and medium-replay.

For the standard offline RL algorithms, to make sure that different types of learning algorithms are covered, we choose state-of-the-art model-based algorithms including MOPO (Yu et al., 2020) and COMBO (Yu et al., 2021), as well as model-free algorithms including IQL Kostrikov et al. (2021) and CQL Kumar et al. (2020). We use OfflineRL-Kit Sun (2023) for the implementation of the model-based algorithms and CORL Tarasov et al. (2024) for the model-free algorithms.

For the baseline methods, to the best of our knowledge, no existing empirical study works in exactly the standard offline PBRL setting considered in our work. The works that consider our setting are either theoretical studies with no empirical implementation (Zhan et al., 2023; Zhu et al., 2023)

| | Oracle | BRL | RM | IPL |
|---|---|---|---|---|
| HalfCheetah M | $76.95 \pm 1.75$ | $\mathbf{75.93 \pm 3.64}$ | $56.52 \pm 0.74$ | $39.71 \pm 2.17$ |
| HalfCheetah MR | $61.24 \pm 3.14$ | $\mathbf{61.72 \pm 0.6}$ | $57.36 \pm 1.6$ | $21.14 \pm 1.14$ |
| HalfCheetah ME | $88.49 \pm 2.61$ | $\mathbf{92.17 \pm 1.67}$ | $87.52 \pm 3.98$ | $34.91 \pm 0.38$ |
| Hopper M | $58.69 \pm 4.67$ | $36.39 \pm 3.9$ | $43.38 \pm 1.0$ | $\mathbf{51.15 \pm 11.89}$ |
| Hopper MR | $76.07 \pm 7.52$ | $\mathbf{28.5 \pm 10.29}$ | $\mathbf{24.44 \pm 3.8}$ | $7.48 \pm 0.42$ |
| Hopper ME | $106.37 \pm 3.73$ | $\mathbf{97.57 \pm 15.08}$ | $74.91 \pm 13.68$ | $28.44 \pm 15.56$ |
| Walker2d M | $76.6 \pm 2.99$ | $49.7 \pm 11.06$ | $58.55 \pm 4.46$ | $\mathbf{67.33 \pm 7.66}$ |
| Walker2d MR | $42.0 \pm 26.95$ | $\mathbf{34.94 \pm 12.21}$ | $13.26 \pm 9.11$ | $14.87 \pm 3.97$ |
| Walker2d ME | $87.54 \pm 40.11$ | $\mathbf{109.94 \pm 1.76}$ | $90.1 \pm 35.01$ | $85.76 \pm 10.43$ |
| Sum Totals: | $673.95$ | $\mathbf{586.86}$ | $506.04$ | $350.79$ |

Table 1: Performance of different learning algorithms on the dataset without overlapped trajectories. 'M' represents the medium dataset, 'MR' represents the medium replay dataset, and 'ME' represents the medium expert dataset.

or empirical studies focusing on fine-tuning LLM (Ouyang et al., 2022; Rafailov et al., 2024) that cannot be applied to general RL settings. We adopt the IPL algorithm from Hejna & Sadigh (2024) as the existing SOTA method among related works. Different from our method, the IPL algorithm requires access to a preference dataset and a behavior dataset. To ensure that the efficiency of IPL is not underestimated, we allow the algorithm to work with the same preference dataset as our method uses and the whole original D4RL dataset with no reward labels as the behavior dataset, which is strictly more information than our method uses. Li et al. (2024) even point out that one can gain high learning efficiency from access to only the D4RL behavior dataset. In addition, we choose the basic reward modeling method (RM) as a natural baseline. The method first learns a reward model from the preference dataset and then labels the dataset with the reward model. Next, any standard RL algorithms can be applied afterward. This method is similar to the standard pipeline in PBRL Ouyang et al. (2022), where the first step is reward modeling, and the second step is RL.

We choose the popular oracle widely considered in PBRL studies (Hejna & Sadigh, 2024; Kim et al., 2023) where a standard offline RL algorithm is used to train on the dataset with true rewards from the RL environment. The dataset contains the same state actions as the preference-based dataset.

## 5.2 Learning Efficiency Evaluation without Trajectory Overlap

Here, we study the most practical case where there is no overlap between trajectories, and each state-action is unique. To straightforwardly compare the performance of different learning methods, we represent the learning efficiency of an algorithm by the performance of the policy learned at the last epoch. In the Appendix, we show the full training log with different learning methods on different datasets. The scores in Table 1 is the standard D4RL score of the learned policy. We observe that our method is better than other baseline methods in general. Note that IPL has additional access to the whole D4RL behavior dataset, which is an advantage compared to our method. In many cases, the learning result of our method can compete with the oracle that trains on the dataset with true reward. We believe our method can sometimes compete with the oracle because the reward dataset already contains more than enough information. As a result, even if the preference dataset contains relatively less information, it is enough for an efficient learner to find a high-performing policy. Our method is, in general, more efficient than the RM baseline. We believe the reason is that our optimal rewards contain more accurate information than the reward model's output with respect to the preference signals.

## 5.3 Overlapped Trajectories

Here, we empirically investigate the case where the trajectory clips in the dataset share the same state-actions. Recall that when there is overlap between trajectories (state-actions in the dataset are not unique), the binary labeling strategy is no longer optimal. The reward modeling is approximating the optimal labels, which can be a stronger baseline in this case. Therefore, in this subsection, we focus on comparing which reward labeling method can give more informative rewards leading

to higher learning efficiency. Specifically, we consider two typical structures of overlap between trajectories.

**I: Same trajectory compared to multiple different trajectories.** In this case, we study a more practical scenario of overlapped trajectories: the same trajectory clip is compared with multiple trajectory clips. More specifically, given a pool of trajectory clips, we sample a portion of them and compare it with multiple different clips from the pool. We set the portion of number of comparisons from low to high to cover a wide range of the degree of trajectory overlap.

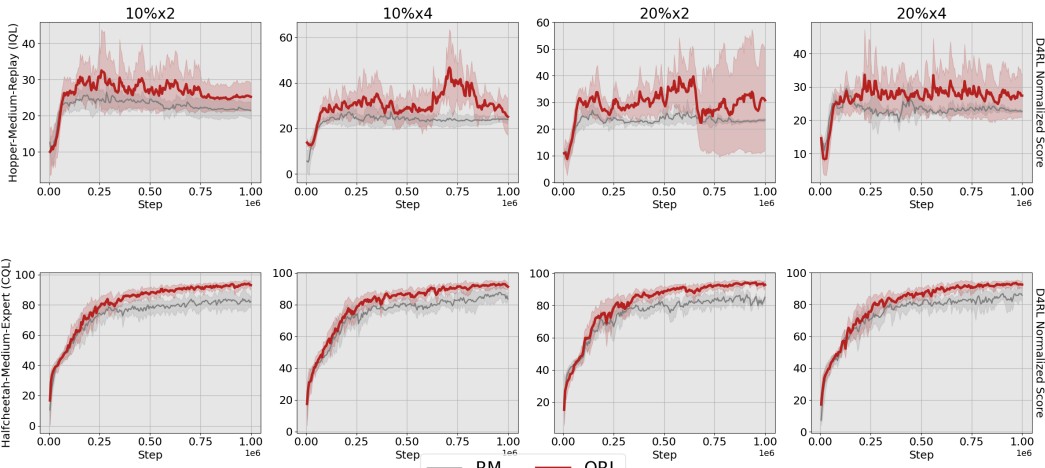

Figure 2: Training log of learning with a method on datasets with overlapped trajectories. The percentage is the portion of trajectories clips that are compared for multiple times compared to all clips. The multiplier is the number of times of multiple comparison. To understand the degree of overlap, in the case of $20\% \times 4$, $80\%$ of the trajectories pairs have a trajectory clip that is compared for multiple times.

The results in Figure 2 show that the BRL method is more efficient than the reward modeling. To understand the reason behind, we check the gap between the reward labels given by BRL and reward modeling. We find that the gap is much larger in the BRL method compared to the reward modeling method, indicating that the binary reward labels keep more information during the feedback signal transition from preference to scalar rewards. The exact comparison results can be found in the Appendix. This could be the reason why BRL method is more efficient.

**II: The same trajectory pair compared multiple times.** Here, we investigate the scenario where multiple preference signals are given to the same pair of trajectory clips. This could happen if multiple users are asked to provide preferences on the same trajectory pairs, and different users may have different judgments. In this case, the preference labels represent an empirical probability of one trajectory being preferred over the other one. Correspondingly, the loss for predicting the probability given reward labels is given by: $|\bar{p} - f(\sum_t r^i_{\sigma^i,t} - \sum_t r^i_{\bar{\sigma}^i,t})|$ where $\bar{p}$ is the empirical probability. We add a regularization term to the loss to ensure the uniqueness of the optimal reward labels. In this case, finding the optimal rewards requires the knowledge of the link function from rewards to preferences. In experiments, we construct the dataset using the same process as before, except that we generate 10 preference labels for each pair of trajectories.

The results in Figure 3 show that in the two datasets we test with, the BRL method is more efficient or as efficient as the reward modeling method. In conclusion, compared to the standard reward modeling method, with or without overlaps between trajectories, the BRL method generally performs better than the reward modeling method. In addition, the BRL method requires no training for function approximation, which is more computationally efficient.

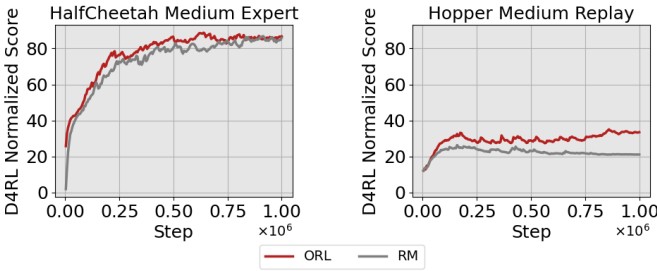

Figure 3: Training log of learning with a method on datasets where 10 preference labels are given to each trajectory pair.

## 5.4 ABLATION STUDY

**Different size of preference dataset:** Here, we examine the efficiency of different methods utilizing the preference signals. For this purpose, we run the algorithms on preference datasets of different sizes. If an algorithm utilizes the preference signals efficiently, then its performance can increase significantly as the number of preference signals increases. We observe in Fig 4 that the learning efficiency of our method increases significantly as the number of preferences increases. In comparison, the learning efficiency of IPL and RM are almost the same for datasets of very different numbers of preference signals. The oracle also has similar performance on the HalfCheetah medium-expert dataset of different sizes. This is likely due to the fact that it always achieves maximal learning efficiency on the dataset.

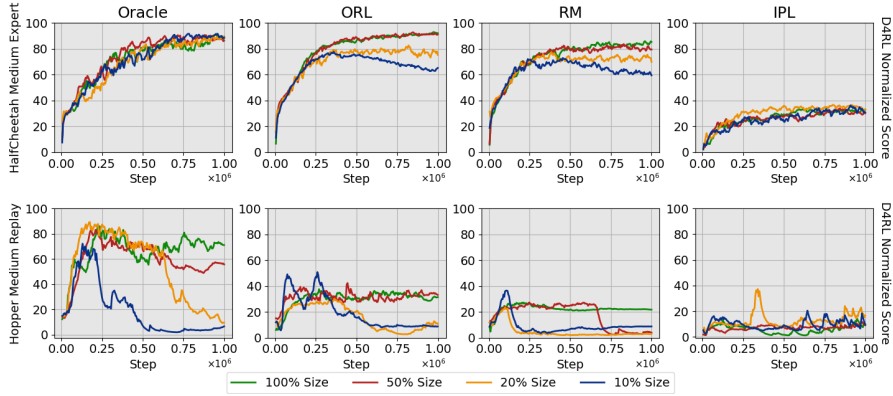

Figure 4: Training log of learning with a method on datasets of different sizes. The percentage in the legends represents the ratio between the number of preference labels in the corresponding dataset and that of the largest dataset.

In the appendix, we show the learning efficiency of combing BRL with different standard Offline RL Algorithms.

## 6 CONCLUSION AND LIMITATION

In this work, we propose a framework BRL that bridges the gap between offline PBRL and standard RL. We show that one can easily achieve high learning efficiency on PBRL problems by combining BRL with any efficient standard offline RL algorithm. Our framework is limited to the typical offline learning setting and does not answer the question of which trajectories are more worthy of receiving preference labels. Our experimental evaluation is limited to continuous control problems, the standard benchmark in RL studies.

## 7 REPRODUCIBILITY

In the main paper, we explain the setting of the problem we study. The proofs for all theorems and lemmas can be found in the appendix. The codes we use for the experiments can be found in the supplementary materials.

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

# A COMBINING BRL WITH MODEL-FREE STANDARD OFFLINE RL ALGORTIHMS

It is typical for a model-free RL algorithm to learn the Q function of the environment (Fujimoto & Gu, 2021; Cheng et al., 2022). The Q function represents the long-term rewards for choosing an action at a state. In the reward-based setting, the Q function is learned with 'Bellman Loss' based on the reward signals. Bellman loss acts as a criterion for the quality of the $Q$ function. Given a tuple $(s, a, r, s')$, the Bellman loss on the data tuple is defined as $|Q(s, a) - (r + \gamma \cdot \max_{a'} Q(s', a')|$. Formally, in Definition A.1, we characterize the Q-learning process in a typical model-free method.

**Definition A.1.** (Q-learning on reward signals in a general model-free RL method with binary reward labels) Given a preference dataset $\mathcal{D} = \{(\tau_1^i \succ \tau_2^i)\}, i \in [N]$. Let $\mathcal{D}_\mathcal{R}$ be the corresponding reward dataset with binary reward labels given by Alg 1. The Q-learning process for a general model-free RL method is solving the optimization problem below:

$$\min_{\hat{Q}} \sum_{i \in [N]} \Big( \sum_{(s,a,s') \in \tau_1^i} |\hat{Q}(s,a) - (1 + \gamma \cdot \max_{a'} \hat{Q}(s', a'))| + \sum_{(s,a,s') \in \tau_2^i} |\hat{Q}(s,a) - (-1 + \gamma \cdot \max_{a'} \hat{Q}(s', a'))| \Big).$$

For simplicity, we denote the optimization goal as $\min_{\hat{Q}} \sum_{i \in [N]} \mathcal{L}_3(\tau_1^i, \tau_2^i, \hat{Q})$.

If one wants to modify a model-free reward-based algorithm to work with preference signals directly, a straightforward way is to replace the Bellman error with the prediction error of the $Q$ function on the preference signals. Based on this method, Hejna & Sadigh (2024) developed the IPL algorithm for PBRL by modifying the IQL algorithm (Kostrikov et al., 2021), a famous method for the standard RL problem. Specifically, the underlying reward function for a $Q$ function is given by $\hat{r}(s,a) = \hat{Q}(s,a) - \gamma \cdot \max \hat{R}(s',a')$. This underlying reward function can be used to predict the probability that the chosen trajectory is preferred to represent the prediction of the $Q$ function. To reflect that the reward function is bounded, we require $Q(s,a) - \gamma \cdot Q(s',a) \in [-1, 1]$ for all $(s, a, s') \sim \mathcal{D}$ so that the corresponding reward function of $Q$ is bounded. Formally, in Definition A.2 we show how a method to perform Q-learning directly on preference signals.

**Definition A.2.** (Q-learning on preference signals through a modified Bellman-Loss) Given a preference dataset $\mathcal{D} = \{(\tau_1^i \succ \tau_2^i)\}, i \in [N]$, the Q-learning on preference signals through a modified Bellman-Loss is solving the optimization problem below:

$$\min_{\hat{Q}} \sum_{i \in [N]} F\Big( \sum_{(s,a,s') \in \tau_1^i} (\hat{Q}(s,a) - \gamma \cdot \max_{a'} \hat{Q}(s',a))) - \sum_{(s,a,s') \in \tau_2^i} (\hat{Q}(s,a) - \gamma \cdot \max_{a'} \hat{Q}(s',a)) \Big).$$

For simplicity, we denote the optimization goal as $\min_{\hat{Q}} \sum_{i \in [N]} \mathcal{L}_4(\tau_1^i, \tau_2^i, \hat{Q})$.

The two methods are closely connected in the optimization problem they solve, and we formally show their connection in three cases in Theorem A.3.

**Theorem A.3.** *Given a preference dataset $\mathcal{D} = \{(\tau_1^i \succ \tau_2^i)\}, i \in [N]$, q-learning can be performed either on the preference dataset directly as Definition A.2 or on the reward dataset as Definition A.1 with the dataset generated binary reward labeling through Alg 1. The two methods are connected in the following three cases:*

1. *When there is no overlap between the trajectories in the dataset, the optimal solutions in both methods are the same:* $\arg\min_{\hat{Q}} \sum_{i \in [N]} \mathcal{L}_3(\tau_1^i, \tau_2^i, \hat{Q}) = \arg\min_{\hat{Q}} \sum_{i \in [N]} \mathcal{L}_4(\tau_1^i, \tau_2^i, \hat{Q})$

2. *If the link-loss function $\mathcal{F}$ is linear, then the optimization problems in both methods are equivalent:* $\sum_{i \in [N]} \mathcal{L}_3(\tau_1^i, \tau_2^i, \hat{Q}) = C_1 \cdot \sum_{i \in [N]} \mathcal{L}_4(\tau_1^i, \tau_2^i, \hat{Q}) + C_2$, where $C_1, C_2$ are constant scalars.

3. *Let $w$ be the parameter of the reward function $\mathcal{R}$. For each trajectory pair, the gradients of its contribution to the optimization goal on the reward function parameter have the same direction in the two methods:* $\frac{\partial \mathcal{L}_3(\tau_1^i, \tau_2^i, \hat{Q})}{\partial w} / \|\frac{\partial \mathcal{L}_3(\tau_1^i, \tau_2^i, \hat{Q})}{\partial w}\| = \frac{\partial \mathcal{L}_4(\tau_1^i, \tau_2^i, \hat{Q})}{\partial w} / \|\frac{\partial \mathcal{L}_4(\tau_1^i, \tau_2^i, \hat{Q})}{\partial w}\|$.

## B  PROOF

### B.1  PROOF FOR LEMMA 4.2

Denote the prediction loss, which is the goal of optimization, as $G = \sum_{i \in [N]} F(\sum_{t \in [T]} r^i_{\sigma^i,t} - \sum_{t \in [T]} r^i_{\bar{\sigma}^i,t}))$. For any $i \in [N]$ and $t \in [T]$, we have

$$\frac{\partial G}{\partial r^i_{\sigma^i,t}} = F'(\sum_{t \in [T]} r^i_{\sigma^i,t} - \sum_{t \in [T]} r^i_{\bar{\sigma}^i,t})) < 0, \frac{\partial G}{\partial r^i_{\bar{\sigma}^i,t}} = F'(\sum_{t \in [T]} r^i_{\sigma^i,t} - \sum_{t \in [T]} r^i_{\bar{\sigma}^i,t})) > 0.$$

Therefore, the prediction loss is monotonically decreasing on $r^i_{\sigma^i,t}$ and monotonically increasing on $r^i_{\bar{\sigma}^i,t}$ for all $i \in [N]$ and $t \in [T]$. Given that the rewards are bound in $[-1, 1]$, the optimal reward that achieves minimal prediction loss is $r^i_{\sigma^i,t} = 1$ and $r^i_{\bar{\sigma}^i,t} = -1$ for all $i \in [N]$ and $t \in [T]$.

### B.2  PROOF FOR THEOREM 4.5 AND A.3

In the first case, when there is no overlap between trajectories, by Lemma 4.2, the optimal reward labels are the binary reward labels in Eq 2. For the two reward modeling processes, the optimal solutions are the reward models that output exactly the same binary reward labels. For the two Q-learning processes, the optimal solutions are the Q functions whose corresponding reward models output exactly the same binary reward labels.

In the second case, recall that the rewards functions and the corresponding reward functions of the Q functions are bounded. We can rewrite $\mathcal{L}_1$ and $\mathcal{L}_3$ as

$$\mathcal{L}_1 = 2 \cdot T - \sum_{(s,a) \in \tau^i_1} \widehat{\mathcal{R}}(s,a)| + \sum_{(s,a) \in \tau^i_2} |\widehat{\mathcal{R}}(s,a) + 1|$$

$$\mathcal{L}_3 = 2 \cdot T - \sum_{(s,a,s') \in \tau^i_1} (\hat{Q}(s,a) - \gamma \cdot \max_{a'} \hat{Q}(s',a')) + \sum_{(s,a,s') \in \tau^i_2} (\hat{Q}(s,a) - \gamma \cdot \max_{a'} \hat{Q}(s',a'))$$

(3)

By comparing that with $\mathcal{L}_2$ and $\mathcal{L}_4$, we get the statements in the second case for both theorems.

In the third case, we check the gradients of the optimization goals at each trajectory pair:

$$L_1 = \sum_{(s,a) \in \tau^i_1} |1 - \hat{R}(s,a)| + \sum_{(s,a) \in \tau^i_2} |\hat{R}(s,a) + 1|$$

$$= \sum_{(s,a) \in \tau^i_1} -\hat{R}(s,a) + \sum_{(s,a) \in \tau^i_2} \hat{R}(s,a) + 2 \cdot T$$

(4)

$$\frac{\partial L_1}{\partial w} = \sum_{(s,a) \in \tau^i_1} -\frac{\partial \hat{R}(s,a,w)}{\partial w} + \sum_{(s,a) \in \tau^i_2} \frac{\partial \hat{R}(s,a,w)}{\partial w}$$

$$L_2 = F\Big( \sum_{(s,a) \in \tau^i_1} \hat{R}(s,a) - \sum_{(s,a) \in \tau^i_2} \hat{R}(s,a) \Big)$$

$$\frac{\partial L_2}{\partial w} = F'\Big( \sum_{(s,a) \in \tau^i_1} \hat{R}(s,a) - \sum_{(s,a) \in \tau^i_2} \hat{R}(s,a) \Big)$$

(5)

$$\cdot \Big( \sum_{(s,a) \in \tau^i_1} \frac{\partial \hat{R}(s,a,w)}{\partial w} - \sum_{(s,a) \in \tau^i_2} \frac{\partial \hat{R}(s,a,w)}{\partial w} \Big)$$

$$L_3 = \sum_{(s,a,s')\in\tau_1^i} |\hat{Q}(s,a) - (1 + \gamma \cdot \max_{a'} \hat{Q}(s',a'))|+$$
$$\sum_{(s,a,s')\in\tau_2^i} |\hat{Q}(s,a) - (-1 + \gamma \cdot \max_{a'} \hat{Q}(s',a'))|$$
$$\frac{\partial L_3}{\partial w} = \sum_{(s,a,s')\in\tau_1^i} \frac{\partial \hat{Q}(s,a) - \gamma \cdot \max_{a'} \hat{Q}(s',a')}{\partial w} -$$
$$\sum_{(s,a,s')\in\tau_2^i} \frac{\partial \hat{Q}(s,a) - \gamma \cdot \max_{a'} \hat{Q}(s',a')}{\partial w}$$

$$(6)$$

$$L_4 = F(\sum_{(s,a,s')\in\tau_1^i} (\hat{Q}(s,a) - \gamma \cdot \max_{a'} \hat{Q}(s',a)))-$$
$$\sum_{(s,a,s')\in\tau_2^i} (\hat{Q}(s,a) - \gamma \cdot \max_{a'} \hat{Q}(s',a))$$
$$\frac{\partial L_4}{\partial w} = F'(\sum_{(s,a,s')\in\tau_1^i} (\hat{Q}(s,a) - \gamma \cdot \max_{a'} \hat{Q}(s',a))-$$
$$\sum_{(s,a,s')\in\tau_2^i} (\hat{Q}(s,a) - \gamma \cdot \max_{a'} \hat{Q}(s',a)))$$
$$\cdot(\sum_{(s,a,s')\in\tau_2^i} \frac{\partial \hat{Q}(s,a) - \gamma \cdot \max_{a'} \hat{Q}(s',a)}{\partial w} -$$
$$\sum_{(s,a,s')\in\tau_1^i} \frac{\partial \hat{Q}(s,a) - \gamma \cdot \max_{a'} \hat{Q}(s',a)}{\partial w})$$

$$(7)$$

Comparing the results and utilizing the fact that the loss-link function is monotonically decreasing, we get the statements in the third case from both theorems.

## C  ADDITIONAL EXPERIMENT RESULTS

**Combing BRL with different standard Offline RL Algorithms:** Here, we examine what is the efficiency of our method when combined with different standard offline RL algorithms training on the same dataset. We observe in Figure 5 that the learning efficiency is higher if one combines ORL with CQL when learning on the HalfCheetah medium-expert dataset. In general, the efficacy of our method is high as long as the offline RL is effective when training on the true rewards.

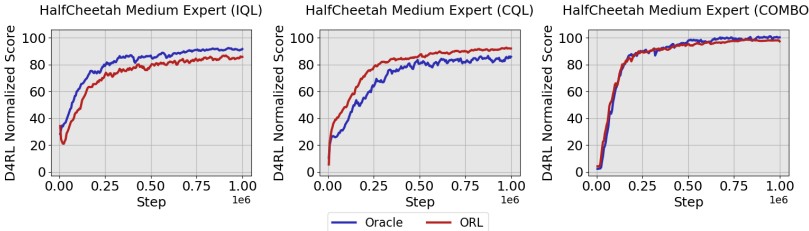

Figure 5: Comparison between the learning efficiency of ORL combined with different standard offline RL algorithms.

In Figure 6, we show the training logs of algorithms learning on different datasets in the main result.

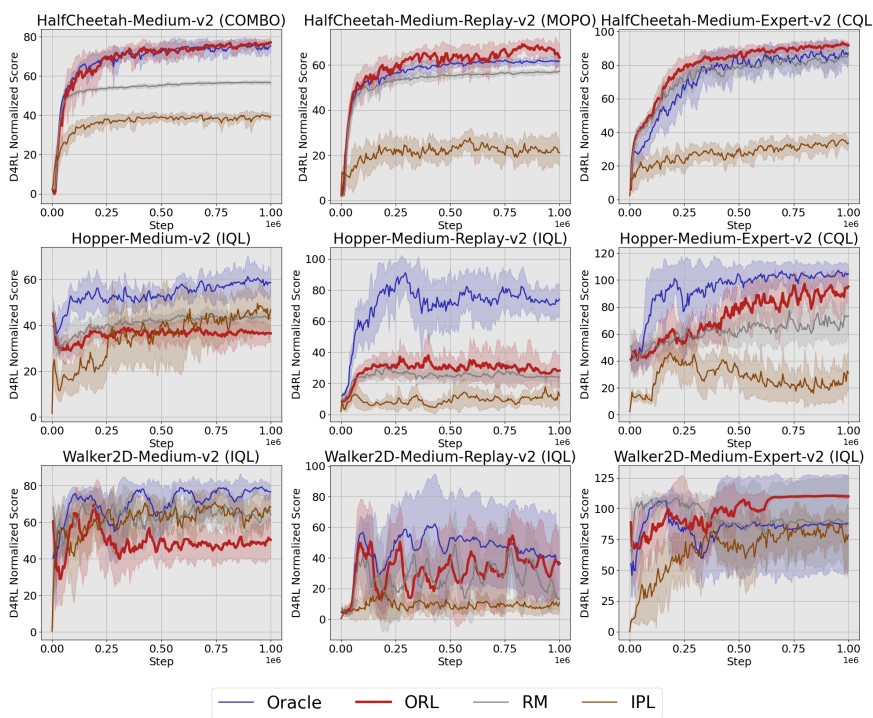

Figure 6: Training log of learning with different methods on different datasets.

| experiment_name | BRL Reward Gap | RM Reward Gap |
|---|---|---|
| HalfCheetah-ME-(10%x2) | 1.930 | 0.420 |
| HalfCheetah-ME-(10%x4) | 1.797 | 0.413 |
| HalfCheetah-ME-(20%x2) | 1.859 | 0.411 |
| HalfCheetah-ME-(20%x4) | 1.581 | 0.420 |

Table 2:

In Table 2, we show the gap between the reward labels given by BRL and RM on the preferred and rejected trajectories. Note that the trajectories in the dataset are overlapped in this case.