# OpenReview forum: "Binary Reward Labeling: Bridging Offline Preference and Reward-Based Reinforcement Learning"
_ICLR.cc/2025/Conference — ICLR 2025 Conference Withdrawn Submission_

### Official Review · Reviewer_Eokm · 2024-10-20

**Soundness:** 3
**Presentation:** 2
**Contribution:** 3
**Rating:** 3
**Confidence:** 3

**Summary:**

This paper proposes a binary-encoding-based reward model learning method for preference-based reinforcement learning. The method demonstrates superior performance in both overlapping and non-overlapping trajectory scenarios.

**Strengths:**

1) Theoretical: In the case of non-overlapping trajectories, the relationship between the binary-encoding-based reward model and the traditional reward model is established.

2) Experimental: The performance of the algorithm is simulated under both overlapping and non-overlapping trajectory scenarios.

**Weaknesses:**

1) Writing: The sections on related work and theoretical foundations are overly redundant. Some statements, particularly in the introduction, are inaccurately expressed. For example, current offline PbRL methods primarily focus on reward model learning, rather than on the policy learning aspect itself. For example, in lines 47-49 and 72-76 of the paper.

2) Motivation: The motivation of the paper is unclear. The authors state that the main goal is to develop a framework to bridge the gap between PbRL and standard RL, allowing a standard offline RL algorithm to address the PbRL problem. However, the primary motivation behind PbRL is to resolve the challenge of setting rewards in standard RL. The difficulty in PbRL lies in accurately learning rewards from human preferences, which is not a problem that standard offline RL addresses. The author could approach this from the perspective of overlapping (or similar) trajectories and inconsistent labels, which might lead to a more effective explanation.

3) Theory: Theoretical 4.5 only considers the case of non-overlapping trajectories and does not account for the scenario of overlapping trajectories with inconsistent labels.

4) Experiments: The dataset is limited, with experiments conducted solely in the mujoco tasks. The paper does not compare results with cutting-edge PbRL methods, such as PT ( Preference transformer: Modeling human preferences using transformers for rl).

**Questions:**

1) Please authors further clarify the motivation of this paper. (This is the main question)

2) How does the algorithm perform in cases where trajectories overlap and labels are inconsistent? The author could discuss how their theoretical results might extend to or be limited by scenarios with overlapping trajectories.

3) What are the advantages of the binary-encoding-based reward model compared to the traditional reward model?

---

### Official Review · Reviewer_KHTZ · 2024-10-22

**Soundness:** 2
**Presentation:** 1
**Contribution:** 2
**Rating:** 3
**Confidence:** 2

**Summary:**

This paper discusses the problem of acquiring a reward function from offline preference datasets. The authors claim that binary reward labelling is sufficient for solving this problem. Results on D4RL demonstrate the effectiveness of the proposed method.

**Strengths:**

1. The problem of reward labeling from preference labels is a fundamental challenge in offline PBRL.
2. The performance improvement is impressive.

**Weaknesses:**

1. Presentation is poor. The citations are poorly formatted and hard to read.
2. Lack of discussion about comparison against the commonly used BT reward model. The contribution is poorly justified.
3. The authors claim that "For the baseline methods, to the best of our knowledge, no existing empirical study works in exactly
the standard offline PBRL setting considered in our work". However, there have been massive studies on offline preference-based RL, such as PreferenceTransformer (https://arxiv.org/pdf/2303.00957) and OPRL (https://arxiv.org/pdf/2301.01392) and can be readily adopted into the experiment framework.
4. (https://proceedings.neurips.cc/paper_files/paper/2023/file/c3e969ea20542a6a11e6caeac736a0b9-Paper-Conference.pdf) reveals that D4RL tasks are not sensitive to reward labels. So the empirical results may not be convincing.

**Questions:**

1. Why does the binary reward outperform BT model? Will the empirical results still hold in more complex tasks such as Meta-World?
2. How do baseline methods such as Preference Transformer perform on the benchmarks?

---

### Official Review · Reviewer_j78D · 2024-11-04

**Soundness:** 2
**Presentation:** 2
**Contribution:** 2
**Rating:** 3
**Confidence:** 4

**Summary:**

This manuscript introduces a novel framework aimed at addressing the challenge of transferring knowledge from reward-based to preference-based offline reinforcement learning (PBRL). The authors highlight that while offline RL has gained practical significance, most research has been limited to scalar reward feedback, leaving a gap in understanding how to apply offline RL techniques to preference-based settings. The proposed solution involves converting preference feedback into scalar rewards through binary reward labeling (BRL), which allows the application of any reward-based offline RL algorithms to datasets with these labels. This approach minimizes information loss during the transition from preference to scalar rewards. The paper establishes theoretical connections between recent PBRL techniques and the proposed framework when combined with specific offline RL algorithms, suggesting that the framework can yield new and more efficient offline PBRL algorithms. Empirical tests on preference datasets from the D4RL benchmark demonstrate that the framework's performance, when combined with various efficient reward-based offline RL algorithms, is often comparable to training on datasets with actual rewards and superior to recent PBRL baselines in most cases.

**Strengths:**

This work investigate an important problem and conduct the theoretical analysis for the method.

**Weaknesses:**

1. The writing of this work is not good. I get confused for many spaces. What is link function? What is link-loss function? The writing of Section 4.1 is very confusing and incomprehensible. The pseudocode is too concise.

2. Missing a lot of baseline algorithms. For example, OPRL [1] and PT [2].

[1] Shin, Daniel, Anca D. Dragan, and Daniel S. Brown. "Benchmarks and algorithms for offline preference-based reward learning." arXiv preprint arXiv:2301.01392 (2023).

[2] Kim, Changyeon, et al. "Preference transformer: Modeling human preferences using transformers for rl." arXiv preprint arXiv:2303.00957 (2023).

**Questions:**

1. Can you evaluate your algorithms on various domains? For example, Antmaze, Kitichen and Adroit?

---

### Official Review · Reviewer_hhtM · 2024-11-04

**Soundness:** 2
**Presentation:** 2
**Contribution:** 2
**Rating:** 5
**Confidence:** 4

**Summary:**

The paper presents a novel framework aimed at bridging the gap between offline preference-based reinforcement learning (PBRL) and standard offline reward-based reinforcement learning (RL). The authors propose a method called Binary Reward Labeling (BRL), which transforms preference feedback into scalar rewards, allowing the application of any reward-based offline RL algorithm to datasets with reward labels. The key insight is simply relabel the reward function with $\pm 1$ using preference labels. The paper provides theoretical connections between PBRL techniques and the proposed framework combined with specific offline RL algorithms. Empirical tests on preference datasets based on the D4RL benchmark demonstrate that the framework's performance is comparable to training on datasets with actual rewards and superior to recent PBRL baselines in many cases.

**Strengths:**

- The paper introduces a simple and unique method for translating preference feedback into a format that can be used by standard offline RL algorithms, which is a significant step forward in the field of PBRL.

- The authors provide a theoretical analysis that connects their framework with existing PBRL techniques, providing an interesting point of view and adding depth to the understanding of how preference information can be utilized in RL.

**Weaknesses:**

- The paper suffers from poor writing quality and formatting issues, which detract from the overall presentation and readability. For example, in Definition 4.2, there should be a period after "reward modeling in model-based approaches," and the comma should not appear at the start of a line. The subtitle "Offline standard RL algorithms are model-based." in Section 4.2 can be misleading.

- The soundness of the proposed method is questionable. While the $\pm 1$ reward labeling is theoretically correct, it is usually not a good choice to overfit the preference dataset. Having a more rigorous analysis under the function approximation scenario would be nice.

- The paper needs some benchmarks and baselines to validate the effectiveness of the proposed method. For benchmarks, The D4RL benchmark is known to be insensitive to the accuracy of the reward function [1], and adding benchmarks like Meta-World would greatly strengthen the paper. Also, there are some recent works on offline PbRL that have a strong performance, like [2,3], and BRL should be compared with them.


References

[1] Li, Anqi, et al. "Survival instinct in offline reinforcement learning." Advances in neural information processing systems 36 (2024).

[2] Kim, Changyeon, et al. "Preference transformer: Modeling human preferences using transformers for rl." arXiv preprint arXiv:2303.00957 (2023).

[3] Zhang, Zhilong, et al. "Flow to better: Offline preference-based reinforcement learning via preferred trajectory generation." The Twelfth International Conference on Learning Representations. 2023.

**Questions:**

See the Weakness section.

---

### Note · Authors · 2024-12-04

**Comment:**

We appreciate the comments from the reviewers and will incorporate them in the next version of our work

**Withdrawal Confirmation:**

I have read and agree with the venue's withdrawal policy on behalf of myself and my co-authors.